# Polyphenols’ Cardioprotective Potential: Review of Rat Fibroblasts as Well as Rat and Human Cardiomyocyte Cell Lines Research

**DOI:** 10.3390/molecules26040774

**Published:** 2021-02-03

**Authors:** Michał Otręba, Leon Kośmider, Anna Rzepecka-Stojko

**Affiliations:** 1Department of Drug Technology, Faculty of Pharmaceutical Sciences in Sosnowiec, Medical University of Silesia in Katowice, Jednosci 8, 41-200 Sosnowiec, Poland; annastojko@sum.edu.pl; 2Department of General and Inorganic Chemistry, Faculty of Pharmaceutical Sciences in Sosnowiec, Medical University of Silesia in Katowice, Jagiellonska 4, 41-200 Sosnowiec, Poland; leon.kosmider@gmail.com

**Keywords:** polyphenols, human and rat cardiomyocytes, rat fibroblasts, cardioprotective activity

## Abstract

According to the World Health Organization, cardiovascular diseases are responsible for 31% of global deaths. A reduction in mortality can be achieved by promoting a healthy lifestyle, developing prevention strategies, and developing new therapies. Polyphenols are present in food and drinks such as tea, cocoa, fruits, berries, and vegetables. These compounds have strong antioxidative properties, which might have a cardioprotective effect. The aim of this paper is to examine the potential of polyphenols in cardioprotective use based on in vitro human and rat cardiomyocytes as well as fibroblast research. Based on the papers discussed in this review, polyphenols have the potential for cardioprotective use due to their multilevel points of action which include, among others, anti-inflammatory, antioxidant, antithrombotic, and vasodilatory. Polyphenols may have potential use in new and effective preventions or therapies for cardiovascular diseases, yet more clinical studies are needed.

## 1. Introduction

According to the World Health Organization (WHO) 2016 datasheet, 17.9 million deaths were caused by cardiovascular diseases (coronary heart disease, cerebrovascular disease, peripheral arterial disease, rheumatic heart disease, congenital heart disease, deep vein thrombosis, and pulmonary embolism), representing 31% of global deaths [1]. The statistics of the European Hearth Network showed that, each year, cardiovascular diseases are responsible for 3.9 million deaths in Europe and over 1.8 million deaths in the European Union (EU), which makes 45 and 37% of deaths in Europe and the EU, respectively. Moreover, the data suggest that the ischemic heart disease and stroke rate is higher in Central and Eastern Europe than in Northern, Southern, and Western Europe [2]. 

In nature, polyphenols play an important role in maintaining fruit color [3], protection against pathogens [3], metal chelating [3], scavenging free radicals [4] as well as antioxidant enzyme inhibition [3]. Polyphenols are classified into two groups: flavonoids and nonflavonoids, which contain sub-classes [5]. The details of polyphenol classification as well as their chemical structures were described in detail in other reviews [5,6,7,8]. Polyphenols may be divided into flavonoid and nonflavonoid groups (see Figure 1) based on their chemical structure. Resveratrol, pterostilbene, piceatannol, and gnetol belong to the stilbenoids, the sub-group of nonflavonoids [9]. 

Polyphenols have gained interest due to their cardioprotective effect, which is related to antioxidative activity [10,11,12,13,14,15], anti-inflammatory activity [11,16,17,18], modulation and synchronization of cardiomyocyte beating [10], and protection against cell apoptosis [11,13,14,19,20,21,22,23,24,25], cardiac hypertrophy [11,21,25,26], and atherosclerosis ischemia/reperfusion injuries [4,11,12,13,15,18]. 

In this review, we focused on the cardiomyocyte and fibroblast assays testing the cardioprotective activity of different polyphenols. Thus, the analyzed studies were mainly based on rat fibroblasts as well as rat and human cardiomyocytes. Some in vivo studies were analyzed.

## 2. Cardioprotective Activity of Polyphenols

### 2.1. Flavonoids

The impact of the three human main polyphenol metabolites (catechol-*O*-sulphate, pyrogallol-*O*-sulphate, and 1-methylpyrogallol-*O*-sulphate) present in berries on rat ventricular neonatal cardiomyocytes and differentiated cardiomyoblast H9c2 cells was analyzed by Dias-Pedroso et al. (2019). To test inhibition of cell death, cells were pre-treated with polyphenol metabolites for 2 h and then incubated with the cardiac-specific cell death inducer-isoproterenol (200 μM for 48h towards H9c2 cells and 24 h and 48 h towards rat cells).

The authors noticed cell death induction in H9c2 and neonatal cardiomyocytes after 48h incubation with isoproterenol, with no protective effect against cell death of their metabolites [10]. The authors also measured the effect of the metabolites on cardiomyocyte cell beating and its synchronization. A 24 h treatment with isoproterenol caused a decrease in the average beating rate, larger distribution of cell beating population, and asynchronous behavior of cardiomyocytes. Noteworthy, 2 h incubation with the metabolites reversed the harmful effect of isoproterenol, causing an increase in the cell beating rate to the level compared with control cells and a narrower cell beating distribution. The reactive oxygen species (ROS) analysis showed that the 24 h incubation of rat cells with isoproterenol stimulates mitochondrial superoxide anion production, leading to cellular stress. A decrease in superoxide anion was observed after pre-treatment of cells with the metabolites, which confirms that the metabolites may decrease oxidative stress caused by isoproterenol treatment [10]. The authors also analyzed if the regulation of cardiomyocyte beating by the metabolites is possible via the cyclic adenosine 3′,5′-monophosphate (cAMP)-dependent protein kinase A/cyclic adenosine monophosphate (PKA/cAMP) pathway and calcium calmodulin-dependent kinase II (CaMKII) kinase. An increase in PKA phosphorylation levels was observed after the incubation of cardiomyocytes with isoproterenol, while pre-treatment with catechol-*O*-sulphate significantly decreased levels of PKA phosphorylation in comparison to the control and isoproterenol treatment group. Moreover, the metabolite caused a significant decrease in the phosphorylation and activation of CaMKII in comparison to the isoproterenol group. Noteworthy, an increase in CaMKII activation was observed after the metabolite treatment in comparison to the control. This suggests that the metabolites may directly stimulate CaMKII in cardiomyocytes in a PKA-independent way and confirm the observed asynchrony cell beating phenotype. During the exposition of cells on isoproterenol, the metabolite directly modulates Ca^2+^ signaling pathways [10].

Chen et al. (2013) analyzed the cardioprotective effect of quercetin (1 mM) in H9c2 rat cardiomyocytes under ischemia/reperfusion injuries. The authors observed that 5 mM H_2_O_2_ treatment led to the strong response of phosphotyrosine, while after the treatment with 1 mM of quercetin and 5 mM H_2_O_2_, the enzymatic response was lower. Moreover, quercetin reduced the tyrosine phosphorylation of Src kinase and focal adhesion kinase (FAK). The wound-healing assay showed that quercetin and H_2_O_2_ treated cells had larger closed areas than H_2_O_2_ only. Similar results were shown by an adhesion assay, suggesting that quercetin stimulates cell migration and maintains cell adhesion. The authors also observed that quercetin significantly reduces the phosphorylation of STAT3, reduces the level of COX-2, inhibits ROS production, and increases the expression of MnSOD in H_2_O_2_-treated cells. This suggests that quercetin may suppress the inflammation process and possess antioxidant activity in H_2_O_2_-induced H9c2 cells. The analysis of apoptosis showed that quercetin reduces apoptosis in cardiomyocytes, as well as decreases Bax and caspase 9 levels and increases the Bcl-2 level [18].

The impact of quercetin on xanthine/xanthine oxidase-induced toxicity in H9c2 cells was analyzed by Özbek et al. (2015). The authors showed that the preincubation of cardiomyocytes with quercetin (0.1–10 μg/mL, 24 h) completely inhibited the production of ROS generated by the xanthine/xanthine oxidase. Moreover, an increase in cellular viability after preincubation was observed. The inhibition of MAPK-2 phosphorylation and ERK1/2 level was also noticed. Quercetin also reversed xanthine/xanthine oxidase-induced down-regulation of phosphorylated heat shock protein 27 (p-Hsp27), p-c-Jun down regulation and inhibited a xanthine/xanthine oxidase-mediated increase in cleaved caspase-3 [19].

### 2.2. Nonflavonoids

The effect of curcumin (8 µM) on gelatinase B (MMP-9) in H9c2 cells was analyzed by Kohli et al. (2013). The authors observed that curcumin prevents norepinephrine-induced hypertrophic stress in H9c2 cardiomyocytes. Moreover, cells treated with curcumin in the presence of norepinephrine prevent an increase in collagen amount, especially collagen-IV involved in heterotrophy, observed after norepinephrine treatment only. A reduction in gelatinolytic activity, mRNA expression, and the protein level of MMP-9 after curcumin treatment was also observed. Finally, the entry of NF-κβ inside the nucleus was also reduced after curcumin treatment—the protein was mainly localized in the cytoplasm [27].

The impact of stilbenoid polyphenols on rat ventricular myocytes was analyzed by Akinwumi et al. (2017). The high concentrations for gnetol (50 and 100 μg/mL) and pterostilbene (5, 10, and 50 μg/mL) significantly decreased cellular viability, while in the lower concentrations the effect was not observed. Moreover, gnetol in the concentration range from 50 to 100 µg/mL significantly reduced cell size in the presence and absence of the second marker of hypertrophy—endothelin 1 (ET1). Interestingly, gnetol in the concentrations 1–10 µg/mL “abolished ET1-induced myocyte enlargement, but did not affect untreated myocytes”. This suggests toxicity rather than anti-growth effect, as well as suggesting that gnetol (5 μg/mL) and pterostilbene (1 μg/mL) possess anti-hypertrophy activity. Gnetol also did not affect AMP-activated protein kinase (AMPK) level, while pterostilbene (1 μg/mL) increased AMPKα phosphorylation at Thr172. The in vivo study using spontaneously hypertensive heart failure (SHHF) and Sprague-Dawley (SD) rats showed that stilbenoid treatment did not affect hypertrophy in vivo and the AMPK level, while resveratrol, gnetol, and pterostilbene significantly improved an indicator of diastolic function—left ventricular isovolumic relaxation time (IVRT) [26].

Pterostilbene (3 µM), in contrast to resveratrol, protects cardiomyocytes against hypoxia-reoxygenation injuries by the activation and up-regulation of sirtuin 1 (SIRT1), which improves mitochondrial function and oxidative capacity. Interestingly, splitomycin as a SIRT1 inhibitor suppresses the protective effects of pterostilbene in H9c2 cells [11]. In vivo studies showed that pterostilbene improved cardiac function and decreased oxidative stress and inflammation markers (tumor necrosis factor α TNFα, interleukin 1β IL-1β, myeloperoxidase activity) in a rat model of ischemia-reperfusion injuries. Moreover, pterostilbene also increases Bcl-2 and decreases Bax expression. Thus, it can protect against apoptosis and reduce cardiac inflammation [11]. Pterostilbene also protects vascular endothelial cells against autophagy by the stimulation of calcium/calmodulin-dependent protein kinase beta, which activates AMPK and finally inhibits the mechanistic target of rapamycin (mTOR) signaling [11]. 

The effect of imine stilbene analog (10 µM) on cardiac hypertrophy and hydrogen peroxide-induced apoptosis of rat H9c2 cells were analyzed by Raut et al. (2020). The authors showed that imine stilbene analogs indicate an anti-hypertrophic effect. The analysis of MAPk kinases involved in pathological cardiac hypertrophy, ERK1/2 (extracellular signal-regulated kinase), JNK1/2 (c-Jun *N*-terminal kinase), and p38 (Mitogen-Activated Protein kinase), showed that imine analogs prevent cardiac hypertrophy. This is possible since a reduction in the ratios of p-ERK/T-ERK, p-JNK/T-JNK, and p-P38/T-P38 was observed. Moreover, the down-regulation of the GATA4 transcription factor and a reduction in p-GSK3β, IGF1R, and PI3K expression were observed after imine analogs treatment. Interestingly, GSK3β is a negative regulator of cardiac hypertrophy, which can be inactivated by phosphorylation. Imine stilbene analog also prevents H_2_O_2_-induced ROS production, reduces expression of caspase 3 and LC3 as well as increases expression of Bcl-2 and AMPK. This suggests the protection of cardiomyocytes against oxidative stress as well as the suppression of autophagy and apoptosis [25].

Neonatal rat cardiomyocytes exposed to 2 h simulated ischemia and 4 h simulated reperfusion treated with resveratrol (100 µM) lead to the attenuation of ischemia-reperfusion injuries. This is possible by the decrease in intracellular calcium, preventing apoptosis and increasing the activity of superoxide dismutase, modulation of the mitochondrial membrane permeability transition pore (mPTP), activation of adenosine monophosphate (AMP)-activated protein kinase (AMPK), and induction of nitric oxide synthase (NOS) [11]. 

Resveratrol in the concentration of 100 µM prevents cardiomyocytes from hypertrophy by the activation of NO-AMPK signaling. Moreover, pressure overload-induced hypertrophy may be reduced by resveratrol via a decrease in oxidative stress and upregulation of endothelial nitric oxide synthase (eNOS), leading to an increase in NO production [11]. Noteworthy, the in vivo study of gnetol and pterostilbenes showed the activation of AMPK in isolated neonatal rat cardiomyocytes, leading to ET-1-induced hypertrophy, while another study showed that 2.5 mg/kg/day of gnetol, pterostilbene, and resveratrol treated spontaneously hypertensive heart failure rats for 8 weeks and did not decrease ventricular hypertrophy [11].

The induction of nuclear factor-κβ (NF-κβ) activity in the human AC16 cell line developing features characteristic of cardiac muscle and neonatal rat cardiomyocytes treated by resveratrol was noticed by Palomer et al. (2012). The authors showed that resveratrol (30 µM) increased expression levels of intercellular adhesion molecule-1 (ICAM-1) and TNFα as well as decreased the expression of proinflammatory genes of interleukin 6 (IL-6) and monocyte chemoattractant protein-1 (MCP-1), regardless of the presence of TNFα. Analysis of the resveratrol effect on NF-κβ activity showed that resveratrol treatment upregulated the DNA-binding activity of NF-κβ, which correlated with an enhancement of MCP-1 and IL-6 mRNA levels. The induction of ICAM-1 or TNFα transcription after resveratrol treatment in human cells was not significant. Moreover, resveratrol induced accumulation of p65 in the nucleus and did not affect NF-κβ levels in human cells. In AC16 cells, resveratrol also activates the signal transducer and activator of the transcription 3 (STAT3) signaling pathway involved in cell survival and proliferation, and induces the expression of the anti-apoptotic Bcl-xL gene. Noteworthy, the Bcl-xL gene is a target of STAT3, contributing to cardioprotective activity [17].

The prevention of doxorubicin-induced cardiotoxicity in H9c2 cells after resveratrol treatment was analyzed by Lou et al. (2015). The authors showed that doxycycline decreases cell viability and induces the expression of ER stress-related apoptotic proteins and subsequent cell death. On the other hand, resveratrol did not decrease viability in the concentration range 0–25 μM, while in the concentrations 50 and 75 μM, it significantly decreased cell viability. Resveratrol (25 μM) and doxycycline (5 μM) treatment increased H9c2 cells’ viability from 58.64 ± 3.10% (doxycycline alone) to 78.92 ± 5.48%. Moreover, resveratrol and doxycycline treatment caused a significant decrease in glucose-regulated protein 78 (GRP78) and a slight decrease in C/EBP homologous protein (CHOP) in comparison to the doxycycline group. Since CHOP is a pro-apoptotic protein, and GRP78 is a key mediator of unfolded protein response, this suggests a cytoprotective role. Resveratrol alone did not affect the level of both proteins. Furthermore, doxycycline and resveratrol treatment increased NAD-dependent class III histone deacetylase-Sirt1 protein levels (doxycycline and resveratrol alone also increase SIRT1 levels), while adding known Sirt1 inhibitor (NIC) decreased Sirt1 levels [28].

Zheng et al. (2015) analyzed the impact of age and resveratrol in the concentration of 10 µM on the hypoxia–reperfusion injuries regeneration in human cardiomyocytes. The authors showed that adding resveratrol as the Sirt-1 activator increases the cellular viability of young and old human cardiomyocytes as well as prevents the increased release of the lactate dehydrogenase (LDH) [29].

The impact of resveratrol (10 µM) on myocardial ischemia/reperfusion injuries was analyzed by Yang et al. (2016). The authors showed that resveratrol significantly reduced myocardial dysfunction in the reperfusion period. Moreover, a decrease in the percentage of infarct area and inhibition of CK-MB release was also observed in vivo. The in vitro study using fetal rat cardiomyocyte-derived H9c2 cells showed that resveratrol attenuates cell death, reduces Bax level, decreases the number of apoptotic cells and mitochondrial membrane potential by about 50%, as well as reduces ROS level by the increase in Mn-SOD and CAT levels, in comparison with the infrared (IR) group. Furthermore, in isolated heart tissues, resveratrol significantly up-regulated mRNA and the protein levels of VEGF-B in comparison with the IR group, as well as up-regulated phosphorylated Akt (Ser473) and GSK3β (Ser 9). All of the observations suggest the potential cardioprotective effect of resveratrol [13].

The impact of resveratrol (50 µM) on the voltage-dependent anion channel 1 (VDAC1) in Sprague-Dawley rat cardiomyocytes subjected to anoxia/reoxygenation injuries was analyzed by Tong et al. (2017). The authors showed that resveratrol significantly increased Sirt 1 and VDAC1 expression in cardiomyocytes. Moreover, a decrease in VDAC1 acetylation was noticed in the resveratrol treatment group. Since the deacetylation of VDAC1 was accompanied by SIRT1 increase, as well as the use of Sirt 1 inhibitor attenuating the deacetylation effect of resveratrol to VDAC1, the authors indicated that resveratrol promoted VDAC1 deacetylation via SIRT1. In the resveratrol treatment group, VDAC1 acetylation also increased BCl-2 and decreased Bax as well as decreased the activity of cardiomyocyte injuries biomarkers—LDH and creatine phosphokinase. Moreover, outer mitochondrial membrane permeability was decreased, which prevented the mitochondrial permeability transition pore (mPTP) from opening. Thus, resveratrol pretreatment may reduce the destruction of mitochondria and cardiomyocytes [20].

Aguilar-Alonso et al. (2018) evaluated the oxidative stress in resveratrol (10 mg/kg/day) treated Male Wistar rat cardiomyocytes. The analysis of NO showed that the decreased level of NO in all treated groups, as well as the decrease, was time dependent. After 2 months of treatment by resveratrol, the concentration of NO^•^ in cardiac cells was 18.49%, while after 8 months it was 62.67%. This is very important since NO is a protective vasodilator agent, but it may also be overgenerated as a second messenger in some situations (aging process, heart and brain ischemia, and hypertension). A reduction in lipoperoxidation products and malondialdehyde generation (major lipoperoxidation products) was also observed. No changes in sodium dehydrogenase (SOD) and catalase (CAT) levels were observed in the tested concentration [15].

The impact of resveratrol on hypoxia/reoxygenation injury-induced oxidative stress in rat cardiomyocytes was analyzed by Li et al. (2019). The authors showed that resveratrol (100 µM) protects against hypoxia/reoxygenation injury-induced structural impairment in the cells. Moreover, resveratrol significantly reduced LDH release and the depolarization of ΔΨm by shifting the ratio of JC-1 monomers, and decreased the Bcl2/Bax ratio, caspase 3 activity, and the cell apoptotic rate, which was induced by hypoxia/reoxygenation injuries. Moreover, resveratrol increased Sirt1, which was decreased by hypoxia/reoxygenation injuries [22].

The effect of resveratrol on primary rat adult fibroblasts, myofibroblasts, and cardiomyocytes was analyzed by Louis et al. (2019). The authors showed that resveratrol (30 and 60 µM) significantly reduced the viability of fibroblasts and myofibroblasts, while it did not alter cardiomyocyte viability. Moreover, a decrease in myofibroblast proliferation by 58% was observed after resveratrol treatment, as well as total nucleic acid content in fibroblasts and myofibroblasts. The increase in apoptotic cells with condensed nuclei by 72 and 62% was observed in resveratrol treated fibroblasts and myofibroblasts, respectively. In contrast, resveratrol did not cause apoptosis in cardiomyocytes [23].

#### Phenolic Acids

The impact of methyl gallate on neonatal rat cardiac myocytes was analyzed by Khurana et al. (2014). The viability assay showed that methyl gallate (MG), gallic acid (GA), epigallocatechin gallate (EGCG), and *N*-acetyl cysteine (NAC) did not significantly change the cellular viability of CoCl_2_-stressed cells. In contrast, H_2_O_2_-stressed cells significantly increased viability after NAC, EGCG, and MG treatment. Moreover, in H_2_O_2_-stressed cells a decrease in membrane integrity with membrane damage was observed, whereas in CoCl_2_-stressed cells round shape and dysfunction of the cellular adhesion were noticed. In case of both stressors, NAC, EGCG, and MG significantly reversed observed changes. The antioxidant activity assays showed that MG and NAC significantly reduced ROS generation and preserved mitochondrial potential as well as increased glutathione. The protection of rat cardiomyocytes against late apoptotic events and nuclear damage was shown with MG and NAC. Moreover, the prevention of cleaved caspase-9 accumulation was also observed [14]. 

Özbek et al. (2015) also analyzed the effect of hydroxytyrosol on xanthine/xanthine oxidase-induced toxicity in H9c2 cells. The authors showed that hydroxytyrosol (0.1–10 μg/mL, 24 h) as quercetin completely inhibited the production of ROS increased cellular viability, and inhibited MAPKAPK-2 phosphorylation and ERK1/2 levels. Hydroxytyrosol in contrast to quercitin increased phosphorylation of Hsp27 and inhibited p-c-Jun in xanthine/xanthine oxidase treated cells. Moreover, hydroxytyrosol as quercetin inhibited caspase 3 [19].

The effect of oleuropein aglycone, the main polyphenol present in olives, on neonatal rat ventricular myocytes’ death was analyzed by Miceli et al. (2018). Oleuropein in 100 µM concentration modified cellular oxidative status and promoted autophagy (an increase in Beclin-1 and LC3-II) with no impact on cell viability. This suggests stimulation of the early autophagy. Oleuropein also activates the autophagic flux in the cells, including autophagosome-lysosome fusion and lysosomal degradation. Moreover, oleuropein increased transcription factor EB (TFEB) translocation to the nucleus, leading to autophagy genes transcription as well as increased mRNA expression of Atp6v1, p62, and Lamp1, suggesting early activation of TFEB transcriptional activity. Noteworthy, TFEB is the master regulator of autophagy and lysosomal genes. Cardiomyocytes post-treated by oleuropein also protected cells against mitochondrial alteration and necrotic cell death [30].

The effect of piceatannol, a natural analog of resveratrol, on PI3KAkt-eNOS signaling in H9c2 cells was analyzed by Wang et al. (2019). The authors showed that piceatannol (10, 20, 40 μM) protected cells against peroxidative injuries induced by hydrogen peroxide in a dose-dependent manner. Piceatannol in the concentration of 40 μM restored cell viability above 80% of the control group. Moreover, piceatannol increased the SOD level, inhibited the reduction in mitochondrial membrane potential (MMP) induced by H2O2, decreased the Ca^2+^ level, and significantly reduced creatine kinase and LDH concentrations as well as the intracellular ROS level in cardiomyocytes. This suggests a protective and antioxidant activity. Piceatannol also increased Bcl-2 expression and decreased cytochrome c and caspase 3 levels, which confirms antiapoptotic activity. Additionally, piceatannol up-regulated PI3K, p-Akt, and eNOS and down-regulated iNOS. This suggests that antiapoptotic activity may be regulated by the PI3K-Akt-eNOS signaling pathway [24].

The effect of two of three human main polyphenol metabolites, pyrogallol-*O*-sulphate and 1-methylpyrogallol-*O*-sulphate, on rat ventricular neonatal cardiomyocytes and differentiated cardiomyoblast H9c2 was also analyzed by Dias-Pedroso et al. (2019). The protective effect of the analyzed metabolites against H9c2 and neonatal cardiomyocytes treated with isoproterenol for 48 h was not observed. As in the case of catechol-*O*-sulphate, 2 h incubation with the metabolites increased cell beating rate and narrower cell beating distribution [10]. Interestingly, the incubation of cells only with the metabolites caused a larger beating distribution of cardiomyocytes, while “1-methylpyrogallol-*O*-sulphate decreased cardiomyocytes’ beating to levels close to isoproterenol treated cells”. The observed effects suggest that the metabolites may modulate cell beating directly and only in some cases correct spontaneous beating defects [10]. Pyrogallol-*O*-sulphate and 1-methylpyrogallol-*O*-sulphate decreased superoxide anion, confirming their antioxidative effect. As in the case of catechol-*O*-sulphate, pyrogallol-*O*-sulphate also significantly decreased levels of PKA phosphorylation in comparison to the control and isoproterenol treatment group. Moreover, all three metabolites significantly decreased the phosphorylation and activation of CaMKII in comparison to the isoproterenol group, CaMKII activation in comparison to the control, and directly modulated the Ca^2+^ signaling pathways. Cell exposure to catechol-*O*-sulphate, pyrogallol-*O*-sulphate, and 1-methylpyrogallol-*O*-sulphate regulated cardiomyocytes beating by the modulation of Ca^2+^ signaling pathways, a decrease in the PKA, and CaMKII activation. Directly modulating cell beating is possible since the metabolites altered CaMKII activation levels without isoproterenol treatment [10].

### 2.3. Polyphenol Containing Plant Concentrates, Extracts, and Fractions

The cardioprotective action of grape polyphenols (Fenocor) was analyzed by the in vivo study of Zadnipryany et al. (2017). The analyzed Fenocor concentrate contained up to 82.69 g/dm^3^ of the total polyphenolic agents amount (mainly gallic acid, catechol, epicatechin, quercetin, and procyanidins) in comparison to red wine 0.2–0.5 g/dm^3^. Two groups of male Wistar rats were administered a CoCl_2_ water solution (60 mg/kg for 7 days). One of the groups received treatment with a grape polyphenol water solution (2.5 mL/kg diluted in 0.05 mL of water). The third group contained non-exposed animals as a control group. Transmission-electron-microscopy (TEM) showed that cobalt chloride (CoCl_2_) in the non-treated group caused degenerative and subnecrotic changes in the myocardium as well as mitochondrial damage (loss of mitochondrial critisae, disruption of the membrane, vacuolization, and appearance of dense osmophilic intramitochondrial particles). Hematoxylin–eosin staining in the same group showed a significant decrease in the length (by 20.74%) and area (by 55.63%) of cardiomyocytes, suggesting hypoxic and ischemic damage. In contrast, the Fenocor-treated group showed a tendency for minimization of the amount of myocardial hypoxic damage after the addition of CoCl_2_. The decrease in length by (13.28%) and area (by 18.07%) of cardiomyocytes, in the same group, suggests the cytoprotective effect of Fenocor treatment. Furthermore, no changes in mitochondria were observed [12]. However, the antioxidant cytoprotective effect did not completely compensate for the destructive effect of powerful hypoxic stress—some cardiomyocytes showed vivid edema in the sarcoplasm and numerous vacuoles in areas of myofibril lysis. The immunohistochemical reaction to desmin lets the authors note that myocardial cells contain varying amounts of desmin in cobalt cardiomyopathy. This is possible since desmin participates in cellular integrity, transport, and mechanical and chemical signaling within cardiomyocytes. In areas with early interstitial collagen fibrosis and cell necrosis, the reaction to desmin was reduced, which may be caused by chronic myocardial ischemia. Moreover, it may explain a reduction in intermediate filaments, cytoplasmic organelles, and myofibrils of cardiomyocytes. In the Fenecor-treated group, cardiomyocytes in the left ventricle usually expressed structural desmin fibers, except for damaged cells. Thus, an increase in desmin relative content in some cardiomyocytes was observed. The analysis of oxidative stress showed an increase of 3.64 times of malondialdehyde (MDA) in blood serum in histotoxic hypoxia as well as an increase in the level of carbonyl compounds by 3.17 times in the non-treated group. This suggests the rapid increase in membrane lipids, peroxide oxidation and sarcolemmal proteins oxidative modification by hydroxyl radicals, which may finally lead to irreversible structural changes in cellular and intracellular membranes. The Fenecor-treated group showed a decrease in lipoperoxidation and protein carbonyl contents [12].

The antioxidant and cardioprotective effects of polyphenolic compounds from grape pomace were also analyzed by Balea et al. (2018). The authors showed that the fresh grape pomace of Fetească neagră (IC_50_ = 9.95 μg TE/mL) possesses much stronger antioxidant activity than fermented grape pomace (IC_50_ = 36.99 μg TE/mL) in comparison to Trolox (IC_50_ = 11.18 μg/mL). In the fresh grape pomace, antioxidative potential (tested with DPPH) was correlated with the amounts of tannins, anthocyanin, flavan-3-ol monomers, and stilbene (r = 0.80–0.99), while in the fermented grape pomace it was correlated with proanthocyanidins, prodelphinidins, and cis stilbenes (r = 0.76–0.90).

Interestingly, fresh grape pomace had a much higher total phenolic index and total anthocyanin complex (161.58 ± 3.42 mg catechin equivalent/g d.w. and 184.84 ± 17.13 mg malvidin-3-*O*-glucoside/g d.w., respectively) than fermented grape pomace (114.71 ± 15.86 mg catechin equivalent/g d.w. and 47.67 ± 5.0 mg malvidin-3-*O*-glucoside/g d.w., respectively). The total concentrations of flavan-3-ol monomer and cis-resveratrol were higher (4.71 ± 0.17 mg/g d.w. and 12.27 ± 3.07 µg/g d.w., respectively) in fermented grape pomace than in fresh grape pomace (2.05 ± 0.21 mg/g d.w. and 5.03 ± 1.09 µg/g d.w., respectively). In contrast, the level of trans-resveratrol was much higher in fresh grape pomace (3.22 ± 1.91 µg/g d.w.) than in fermented grape pomace (0.58 ± 1.01 µg/g d.w.) [4]. The in vivo study using rats did not show any significant effect, but on day 10 the authors noticed that the isoprenaline injected group had significantly altered electrocardiogram (ECG) patterns (decrease in heart rate and increase in RR, QT, and QTc intervals). Moreover, the isoprenaline-induced infarct-like lesion was reflected by the depression of ST-segment and marked T wave inversion. Noteworthy, the fresh grape pomace treated group (1 mL/day p.o. for 7 days, administrated orally by gavage) showed a protective effect since a reduction in RR and QT, an increase in QTc intervals, as well as an increase in heart rate and a reduction in ST depression were noticed in comparison to isoprenaline-treated animals. In contrast, the fermented grape pomace treated group only showed a reduction in RR interval, an increase in the heart rate, and a reduction in ST depression in comparison to the isoprenaline treated group. The analysis of serum cardiac marker enzymes showed an increase in aspartate aminotransferase (AST), alanine transaminase (ALT), and creatine kinase myocardial band (CK-MB) levels in the isoprenaline treated group in comparison to the control, while in fresh and fermented grape pomace treated animals, a significant reduction in the analyzed enzymes was observed in comparison to the isoprenaline treated group. Finally, the oxidative stress analysis performed by the authors showed that isoprenaline induced myocardial ischemia rats significantly increased the oxidative stress index and total oxidative status in serum in comparison to the control. In contrast, both fresh and fermented grape pomace treated groups significantly decreased the oxidative stress index and total oxidative status in serum in comparison to isoprenaline induced myocardial ischemia rats. Furthermore, isoprenaline-induced ischemia was associated with a significant increase in malondialdehyde (MDA) nitrites and nitrate (NOx) levels, as well as a decrease in total thiols level (SH) in comparison to the control, while fresh and fermented grape pomace treatment significantly reversed these changes [4].

The impact of the *Acacia hydaspica* extract (400 mg/kg b.w./day for 6 weeks) on oxidative stress and structural alterations in rat cardiomyocytes was analyzed by Afsar et al. (2017). The analysis of cardiac biomarkers (CK, CKMB, AST, and LDH) showed that the extract reduced the toxic effect of doxycycline in a dose-dependent manner. Moreover, analysis of antioxidant enzymes showed that the polyphenol-rich extract (gallic acid, catechin, myrcytein, and rutin) restored the levels of catalase, peroxidase, superoxide dismutase, and quinone reductase, which were significantly decreased by doxycycline treatment only. The prevention of extract co-treated cells against reduced glutathione, glutathione peroxidase, γ-glutamyl transpeptidase, glutathione reductase, and glutathione S-transferase activity depletion was also noticed. Finally, the extract in co-treatment cells prevented the increase in malondialdehyde, H_2_O_2_, and nitrite contents, which was observed in doxycycline only treated cells [31].

The antioxidant effect of polyphenol-rich fraction of citrus bergamot (20 mg/kg daily for 14 days) doxorubicin-induced cardiomyopathy on Wistar male rat cardiomyocytes and cardiac stem cells was analyzed by Carresi et al. (2018). The authors showed that doxorubicin treatment (six doses of 2.5 mg/Kg from day 1 to day 14) increases left ventricular contractility, while bergamot polyphenol fraction (neoeriocitrin, naringin, neohesperidin, bruteridin, melitidin, and flavonoids 6,8-di-*C*-glycosides) treatment significantly prevents it. Moreover, the number of apoptotic cells increased to 1.4 ± 0.3% in the doxorubicin treated group with accompanying hypertrophy, while in the control and bergamot polyphenol fraction treated group the number of apoptotic cells was 0.05 ± 0.01% and 0.04 ± 0.01%, respectively, without hypertrophy. The research based on cardiac stem cells showed that the polyphenol fraction prevented doxorubicin cellular toxicity, while the number of newly generated cells was almost the same (0.015 ± 0.007%, 0.017 ± 0.010%, and 0.014 ± 0.004%) for doxorubicin, bergamot polyphenols, and the control group, respectively. Interestingly, the number of replenishing cardiomyocytes significantly increased up to 0.571 ± 0.074% in doxorubicin and bergamot polyphenol treated cells in comparison to the control, doxorubicin, and bergamot polyphenol fraction only. 

Moreover, bergamot polyphenolic fraction significantly reduced 8-OH-deoxyguanosine (a common DNA adduct resulting from injuries to DNA via ROS), γ-H2AX signal (key DNA damage signal reflecting primarily DNA double-stranded breaks), and ROS formation, and the specific formation of malondialdehyde in cardiomyocytes. It also increased levels of protein nitration, Beclin-1, p62, microtubule-associated protein 1 light chain 3B isoform I (LC3B-I), and LC3B-II expression. This suggests the induction of autophagy since Beclin-1 (autophagy protein marker) plays an important role in the early steps and the formation of the autophagosome, p62 is a marker of substrate sequestration into an autophagosome, and the ratio of LC3B-II/LC3B-I is a marker of autophagic pathway activation by LC3 cleavage. In the case of stem cells, the doxorubicin and bergamot polyphenol fraction treated group significantly reduced the number of 8-OHdG-positives to 31.4 ± 3.3% in comparison to the doxorubicin only treated group [21].

## 3. Discussion and Conclusions 

New and more effective methods of prevention and/or treatment of cardiovascular diseases are important to obtain as an increase in cardiovascular disease caused deaths has been observed globally. Polyphenols, which possess anti-inflammatory, antioxidant, antithrombotic, cardioprotective, and vasodilatory effects, give some hope for future applications, especially as the mentioned effects were confirmed by several groups in in vitro and in vivo studies.

The effect of different polyphenols on fibroblast cardiomyocytes, AC16 [17], myofibroblasts [23], H9c2 [10,11,13,18,19,24,25,27,28], neonatal rat cardiomyocytes [10,11], primary rat adult fibroblasts [23], rat cardiomyocytes [12,15,20,21,22,23,26,30,31], as well as young and old human cardiomyocytes [29] was analyzed by different groups. 

Resveratrol attenuates ischemia-reperfusion injuries in neonatal rat cardiomyocytes by the decrease in intracellular calcium, preventing apoptosis and increasing the activity of superoxide dismutase, modulation of the mPTP, activation of AMPK, and induction of NOS. Moreover, resveratrol prevents cardiomyocytes from cardiac hypertrophy by the activation of NO-AMPK signaling, a decrease in oxidative stress, and upregulation of eNOS, leading to an increase in NO production [11]. In the case of rat cardiomyocytes aging, resveratrol directly decreases antioxidant markers (nitric oxide and total lipoperoxidation) with no influence on antioxidant enzymes, which was confirmed by Aguilar-Alonso et al. (2018) [15]. Noteworthy, resveratrol does not inhibit proliferation and cell death of cardiomyocytes, in contrast to cardiac fibroblasts and myofibroblasts, which was shown by Louis et al. (2019). This suggests that using resveratrol may affect heart function, while the ability of resveratrol to stimulate apoptosis suggests that it may limit the pathological effects of cardiac fibrosis by limiting cardiac fibroblast/myofibroblast expansion. Thus, the study confirms the cardioprotective role of resveratrol as a potential agent used in a new supplementary therapy against cardiac diseases involving cardiac fibrosis [23]. The ability of resveratrol to reduce mitochondrial oxidative stress [13,22], protect against apoptosis [13,22,28], and restore the Sirt1 level [22,28] in cardiomyocytes was also shown by Li et al. (2019), Lou et al. (2015), and Yang et al. (2016). Resveratrol protects from a decrease in the levels of GRP78 and CHOP and cell death after doxycycline treatment. This suggests a protective role of resveratrol, such as a reduction in DOX-induced cardiotoxicity as well as ER stress-induced injuries and homeostasis by the activation of the Sirt1 pathway [28]. The VDAC1 deacetylation by Sirt 1 after treatment of cardiomyocytes by resveratrol was noticed by Tong et al. (2017). Interestingly, the interaction between VDAC1 and Bax increased the binding with Bcl-2 and prevented mPTP opening. This suggests the protection of the cells against apoptosis [20]. Moreover, the ability of resveratrol to increase Sirt-1 may enhance the cardioprotective effect of hypoxia, which was confirmed by Zheng et al. (2015), who noticed increased viability and decreased LDH release [29]. Resveratrol also attenuates the VEGF-B signaling pathway in cardiomyocytes [13] and induces NF-κB activity in human AC16 cells and neonatal rat cardiomyocytes [17]. Moreover, resveratrol activates the STAT3 pathway only in AC16 cells, which induces the expression of anti-apoptotic Bax, suggesting the cardioprotective role of resveratrol [17]. Noteworthy, the cardiac role of VEGF-B is not clear since the overexpression of VEGF-B may protect against acute myocardial infarction, while other studies show that it is not critical to cardiac repair. Thus, resveratrol can be a promising cardioprotective adjuvant in myocardial IR therapy and a modulator for clinical myocardial therapy [13]. 

Quercetin inhibits the tyrosine phosphorylation of Src kinase and FAK, leading to a decrease in cell–cell interaction [18] and morphology [18], apoptosis [18,19], ROS production [18,19] as well as stimulating migration [18], and this affects H9c2 rat cardiomyocyte adhesion [18] and cardiomyocyte viability [19]. Moreover, it also inactivates STAT3 inactivation, which may be beneficial for an ischemia/reperfusion injury model [18], which was shown by Chen et al. (2013) and Özbek et al. (2015). Quercetin and hydroxytyrosol preincubation of cardiomyocytes also protects cardiac hypertrophy, as the most important role in cardiac hypertrophy is played by mitogen-activated protein kinases (MAPKs), including extracellular responsive kinases (ERKs), stress-activated protein kinases (SAPKs) such as c-Jun N-terminal kinases (JNKs), and p38 MAPKs. The inhibition of MAPKAPK-2 phosphorylation by the analyzed polyphenols is also very important since the ROS-mediated chronic activation of p38 MAPK and its downstream cascade have been implicated in myocyte hypertrophy. A similar situation is found with the inhibition of ERK 1 and 2 by polyphenols since uncontrolled activation of Ras-Raf-MEK-ERK signaling triggers hypertrophic cardiomyopathy, confirming the cardioprotective activity of quercetin and hydroxytyrosol [19].

Isoproterenol increases ROS generated arrhythmogenic Ca^2+^ transients and decreases cardiomyocyte beating and synchrony after chronic exposition to the cardiomyocytes. This is possible by the stimulation of the β-adrenergic receptor. On the other hand, 2 h pre-treatment of the cells by the three phenolic metabolites inhibits oxidative stress, improves Ca^2+^ transients by the modulation of amplitude and/or synchronization, and limits PKA/CaMKII phosphorylation. Finally, it may lead to the modulation of cardiomyocyte beating. Noteworthy, the relaxation of cardiomyocytes and heart rate are possible via phosphorylation and activation of PKA, followed by phosphorylation of other key proteins involved in Ca^2+^ dynamics and excitation-contraction coupling. However, the chronic stimulation of these receptors may be harmful and lead to arrhythmogenic cell beating behavior of cardiomyocytes. Thus, these observations confirm cardioprotective and antioxidant activity and suggest that the metabolites directly modulate Ca^2+^ signaling, cardiomyocyte beating and synchrony [10]. 

Bergamot polyphenols also possess a cardioprotective role since they reduce acute anthracycline-induced cardiac toxicity and hypertrophy, activate protective autophagy and increase cardiomyocyte viability, as well as prevent apoptosis and cardiac stem cell damage. Moreover, they protect against oxidative stress by the reduction in 8-OH-deoxyguanosine, γ-H2AX signal, and ROS formation. This suggests that bergamot polyphenols may cause beneficial effects in attenuating cardiotoxicity in patients requiring anthracycline chemotherapy [21].

Curcumin prevents norepinephrine-induced hypertrophic stress and an increase in collagen amount, and reduces gelatinolytic activity, the entry of NF-κB inside the nucleus, mRNA expression, and protein level of MMP-9 in H9c2 cardiomyocytes, which was confirmed by Kohli et al. (2013). All the results suggest that curcumin possesses cardioprotective activity since it may protect against heart hypertrophy [27]. 

Gnetol and pterostilbene impact the hypertrophy of isolated cardiomyocytes via AMPK signaling, which was confirmed by Akinwumi et al. (2017). Interestingly, the study showed the different effects of the analyzed polyphenols between in vivo and in vitro studies [26]. It suggests that we have to be careful in pursuing new therapies for human cardiovascular disease as well as that in vitro studies should be compared with in vivo studies to be sure that the desired effect will be present in the therapy. Moreover, pterostilbene protects H9c2 cardiomyocytes against hypoxia-reoxygenation injuries by the activation and up-regulation of SIRT1 [11].

Methyl gallate increases viability and glutathione levels and reduces ROS. Moreover, methyl gallate protects mitochondria, protects cells against apoptosis, protects DNA from damage, and reduces caspase-9 activation by oxidative stress, which was shown by Khurana et al. (2014). Thus, the cardioprotective role of methyl gallate may be caused by repressing apoptosis protection and the scavenging of free radicals, which suggests therapeutic value in cardioprotection [14]. 

Oleuropein stimulates autophagy and protects cells against mitochondrial dysfunction and cell death, which was noticed by Miceli et al. (2018). Moreover, oleuropein activates TFEB and increases the expression of genes regulating autophagy. Interestingly, TFEB modulation may delay organ degeneration and prevent cardiac disease via restoring autophagy [30].

Piceatannol protects H9c2 cardiomyocytes against oxidative stress and apoptosis, which was shown by Wang et al. (2019). Interestingly, the antiapoptotic activity may be mediated by the PI3K-Akt-eNOS signaling pathway [24]. 

The cardioprotective activity of the imine stilbene analog was noticed by Raut et al. (2020). This is possible since the analog protects cardiomyocytes against oxidative stress, suppresses autophagy and apoptosis, and indicates anti-hypertrophic activity [25]. 

Polyphenol-rich *Acacia hydaspica* extract may be beneficial for doxycycline-induced cardiotoxicity treatment, which was shown by Afsar et al. (2017) [31]. 

The impact of polyphenols on human and rat fibroblasts and cardiomyocytes is shown in Figure 2.

In recent years, papers on cardioprotective polyphenol use in animals [4,11,12,13,26,32,33] and humans [11] have appeared, and their results support the in vivo studies. 

Yang et al. (2016) showed that resveratrol decreases the percentage of infarct area and inhibition of CK-MB release in vivo [13]. The in vivo study of Zadnipryany et al. (2017) showed that polyphenol rich Fenocor concentrates possess antioxidant activity and significantly decrease harmful changes observed in the myocardium stimulated by CoCl_2_. The lack of total prevention of cobalt-induced cardiac damage may be explained by the different mechanism and can affect the cellular redox balance of the cobalt toxic effects [12]. The results obtained by Balea et al. (2018) suggest that grape pomace extracts possess a cardioprotective effect against isoprenaline-induced myocardial ischemia changes by the antioxidant activity. This is possible as the extracts contain polyphenols that have antioxidant activity. Noteworthy, the antioxidant effect depends on the type of grape pomace since fresh grape pomace has better in vivo antioxidant activity, while fermented grape pomace possesses a stronger in vitro antioxidant effect. The results presented by the authors also suggest that the grape pomace extract may be an option for heart preconditioning [4]. Moreover, Akinwumi et al. (2018) suggest the cardioprotective activity of stilbenoids (astringin, gnetol, piceatannol, and pterostilbene) because of the decrease in mortality rate, Bax expression, TGFβ, oxidative stress, and inflammation markers. Additionally, an increase in Bcl-2 and NO was observed in rats and mice in vivo [11]. Noteworthy, the in vivo studies performed by Rzepecka-Stojko et al. (2017 and 2018) confirmed the protective role for cardiac arteries of bee pollen polyphenols. They significantly limit or completely prevent atherosclerosis as well as reduce and/or prevent hepatic steatosis and degenerative changes caused by a high-fat diet [32,33]. Moreover, in vivo as well as human studies confirm the cardioprotective role of polyphenols, as shown in the previous paper [34]. Despite the limited bioavailability of polyphenols [35], studies performed on humans [36,37,38,39,40] confirm the cardioprotective activity of polyphenols in in vivo design. The role of foods rich in polyphenols (pterostilbene, grape, cocoa, and tea polyphenols) in the diet was confirmed by the reduction in systolic and diastolic blood pressure [36], improvement of the inflammatory and the fibrinolytic status of patients [37] and atherothrombotic signaling [38], decreased plasma triglyceride concentration, stimulation of antioxidant activity, inhibition of platelet aggregation, as well as reduced inflammation and biological membrane lipid peroxidation [39,40]. Moreover, the “French paradox” also confirms the cardioprotective activity of polyphenols [41]. The in vitro studies described here may be used to explain the mechanism of action of cardiomyocytes and fibroblasts, but they cannot be referenced directly to the clinical use. Moreover, the described studies can be used to compare previous and new results. Thus, more in vivo research is needed to support that claim.

Based on the collected data, polyphenols possess cardioprotective properties through multidirectional pathways, leading to protection against apoptosis, cardiac hypertrophy, or ischemia/reperfusion injury and others, mostly due to their antioxidative properties. The detailed cardioprotective effects of polyphenols on fibroblasts and cardiomyocytes is summarized in Figure 2. 

## Figures and Tables

**Figure 1 molecules-26-00774-f001:**
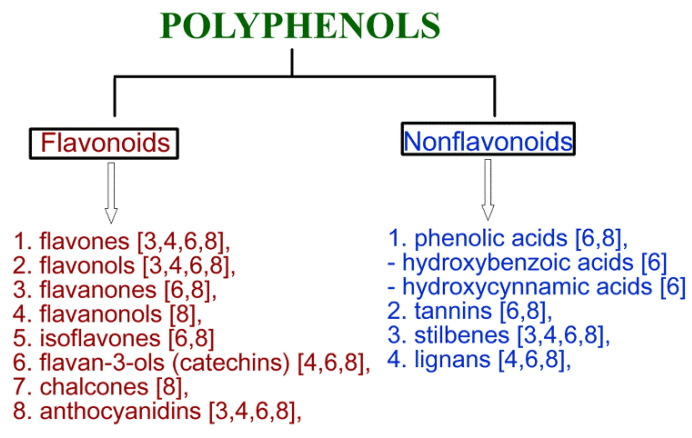
The classification of polyphenols based on their chemical structure.

**Figure 2 molecules-26-00774-f002:**
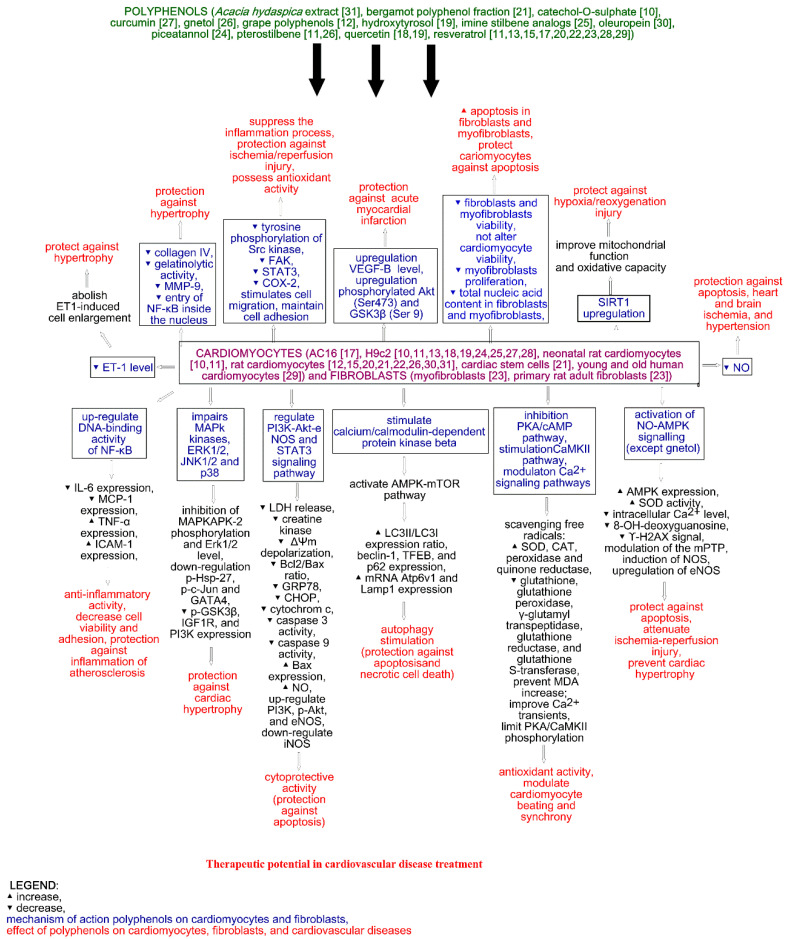
The summary of cardioprotective effects of polyphenols on fibroblasts and cardiomyocytes; Akt—protein kinase B, cAMP cyclic adenosine 3′,5′-monophosphate, CHOP—C/EBP homologous protein, COX-2—cyclooxygenase 2, ERK—Extracellular signal-regulated protein kinase, ET-1—endothelin 1, FAK—focal adhesion kinase, GATA4—GATA Binding Protein 4, GRP78—glucose-regulated protein 78, GSK3β—glycogen synthase kinase 3β, Hsp—Heat shock protein, ICAM-1—intercellular adhesion molecule 1, IGF1R—insulin-like growth factor 1 receptor, JNK—c-Jun *N*-terminal kinases, Lamp1—Lysosomal-associated membrane protein 1, LC3—microtubule-associated protein light chain 3, LDH—Lactate dehydrogenase, MAPK—mitogen-activated protein kinases, MCP-1—monocyte chemoattractant protein-1, MDA—malondialdehyde, MMP-9—matrix metallopeptidase 9, MPTP—1-methyl-4-phenyl-1,2,3,6-tetrahydropyridine, NF-κβ—nuclear factor κβ, NO—nitric oxide, PI3K—phosphoinositide 3-kinases, SIRT1—sirtuin 1, STAT—signal transducer and activator of transcription, TFEB—transcription factor EB, TNF-α—tumor necrosis factor α, VGF-β—vascular endothelial growth factor β.

## Data Availability

No new data were created or analyzed in this study.

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
