# Peer review of "Polyphenols’ Cardioprotective Potential: Review of Rat Fibroblasts as Well as Rat and Human Cardiomyocyte Cell Lines Research"

_molecules, 2021, doi:10.3390/molecules26040774_

Round 1

Reviewer 1 Report

I read the re-submitted manuscript and I'm afraid this review is still not comprehensive enough. As polyphenols has very low bioavailability, its direct effects on the cardiomyocytes are quite limited. Authors summarized the in vitro data, which are difficult for translating into clinical practice.

Author Response

Reviewer 1:

I read the re-submitted manuscript and I'm afraid this review is still not comprehensive enough. As polyphenols has very low bioavailability, its direct effects on the cardiomyocytes are quite limited. Authors summarized the in vitro data, which are difficult for translating into clinical practice.

We would like to thank the reviewer for the suggestions. Please find bellow responses for all recommendations.

We agree with the reviewer that the bioavailability of polyphenols is low, but the number of papers presents the positive cardioprotective effect of those compounds when using on humans and animals. Summary of the in vitro studies can not be directly referenced to clinical practice, they can be used as the basis for clinical trials and explain the mechanism of action. We have added the paragraph in the Discussion and conclusions section:

“Despite the limited bioavailability of polyphenols [35], studies performed on humans [36-40] confirm the cardioprotective activity of polyphenols in in vivo design. The role of foods rich in polyphenols (pterostilbene, grape, cocoa, and tea polyphenols) diet was confirmed by the reduction of systolic and diastolic blood pressure [36], improvement of the inflammatory, the fibrinolytic status of patients [37], and atherothrombotic signaling [38], decrease plasma triglyceride concentration, stimulate antioxidant activity, inhibit platelet aggregation, as well as reduced inflammation and biological membrane lipid peroxidation [39-40]. Moreover, the “French paradox” also confirms the cardioprotective activity of polyphenols [41]. The described here in vitro studies may be used to explain the mechanism of action of cardiomyocytes and fibroblasts but they can not be referenced directly to the clinical use. Moreover, the described studies can be used to compare previous and new results.”

Reviewer 2 Report

The manuscript entitled “Polyphenols cardioprotective potential - review of rat fibroblasts as well as rat & human cardiomyocytes cell lines research” presents a review on the cardioprotective potential of some polyphenolic compounds.

Major revisions should be made in order to be published in Molecules journal, and the manuscript should be completed and/or modified taking into account the suggestions from the attached file.

Author Response

Reviewer 2

Polyphenols cardioprotective potential - review of rat fibroblasts as well as rat & human cardiomyocytes cell lines research cardioprotective potential of some polyphenolic compounds.

We would like to thank the reviewer for the detailed suggestions. Please find bellow responses for all recommendations made the point by point. All changes are also marked in the text using red font.

Major revisions should be made in order to be published in Molecules journal, and the

manuscript should be completed and/or modified taking into account the suggestions below:

  1. The authors are advised to rephrase the sentences from lines: 24-27, 43, 226-227, 261-263, 297-299, 338-341, 453-454; 454-455, 516-517, 533-535, 539-541, 542-544.

Thank you for taking afford to list all sentences which required modification. We have rephrased all sentences in accordance to the list provided.

  1. The authors are advised to use a classification of polyphenolic compounds based on their chemical structure
  2. The authors are advised to correct the term anthocynidins Fig. 1

Thank you for the suggestion, we changed the classification of polyphenolic compounds as asked and now we based it on our previous paper (reference no 8). The Figure 1 was also corrected.

  1. The authors are advised to use the term for instead of in line 117.

The corrections was made

  1. The authors are advised to better explain why they marked with red the lines 422-426,

since the sentence was the same as in previous version of manuscript.

Please find our apologies, this sentence was marked as we moved this sentence.

  1. The authors are advised to correct the terms hydroxytyrosol / hydrocortyzol line

493, 502

The correction was made

  1. The authors are advised to correct the term confirms line 537

Thank you for point out the error. The term was corrected.

  1. The authors are advised to change the Discussion and conclusions section (3). There is a

mix between the papers and the compounds and itt is very difficult to follow. The authors

should present a comparison between the studies for the same compound, in order to have

an overview on the compound.

Thank you for the reviewer's comment. We have rearranged the discussion section with a focus on compounds in in vitro studies so it is easier to follow.

Round 2

Reviewer 2 Report

The authors made all the required changes and the manuscript has been significantly improved.

This manuscript is a resubmission of an earlier submission. The following is a list of the peer review reports and author responses from that submission.

Round 1

Reviewer 1 Report

The manuscript entitled “Polyphenols cardioprotective potential - review of rat fibroblasts as well as rat & human cardiomyocytes cell lines research” presents a review on the cardioprotective potential of some polyphenolic compounds.

Major revisions should be made in order to be published in Molecules journal, and the manuscript should be completed and/or modified taking into account the suggestions from the attached file.

Reviewer 2 Report

The present review described the recent advances regarding the cardioprotective potential by polyphenols. I read this review and I’m afraid this paper did not meet the publishing requirements by Molecules. My reasons are listed below.

Polyphenols can be divided into three groups, flavonoids, nonflavonoids, and phenolic acids.

This definition is partly incorrect. As there are several subgroups, the large parts of polyphenols, like tannins, are missing.  

The figures are poorly drawn. Figure missed key important structures of polyphenols.

Authors claimed that polyphenols has cardioprotective effect, however this is lack of supporting clinical data. Without population data, in vitro cell models are too weak to get significant impact.

Figure 2 contains too many information which makes readers difficult to get key point.

I’m surprising that as a long review authors only cite 27 papers in there reference list.